# Online Learning with an Unknown Fairness Metric

**Stephen Gillen**
University of Pennsylvania
stepe@math.upenn.edu

**Christopher Jung    Michael Kearns    Aaron Roth**
University of Pennsylvania
{chrjung, mkearns, aaroth}@cis.upenn.edu

## Abstract

We consider the problem of online learning in the linear contextual bandits setting, but in which there are also strong *individual fairness* constraints governed by an unknown similarity metric. These constraints demand that we select similar actions or individuals with approximately equal probability [**?**], which may be at odds with optimizing reward, thus modeling settings where profit and social policy are in tension. We assume we learn about an unknown Mahalanobis similarity metric from only weak feedback that identifies fairness violations, but does not quantify their extent. This is intended to represent the interventions of a regulator who "knows unfairness when he sees it" but nevertheless cannot enunciate a quantitative fairness metric over individuals. Our main result is an algorithm in the adversarial context setting that has a number of fairness violations that depends only logarithmically on $T$, while obtaining an optimal $O(\sqrt{T})$ regret bound to the best fair policy.

## 1   Introduction

The last several years have seen an explosion of work studying the problem of fairness in machine learning. Yet there remains little agreement about what "fairness" should mean in different contexts. In broad strokes, the literature can be divided into two families of fairness definitions: those aiming at *group* fairness, and those aiming at *individual* fairness.

Group fairness definitions are aggegrate in nature: they partition individuals into some collection of *protected groups* (say by race or gender), specify some statistic of interest (say, positive classification rate or false positive rate), and then require that a learning algorithm equalize this quantity across the protected groups. On the other hand, individual fairness definitions ask for some constraint that binds on the individual level, rather than only over averages of people. Often, these constraints have the semantics that "similar people should be treated similarly" **?**.

Individual fairness definitions have substantially stronger semantics and demands than group definitions of fairness. For example, **?** lay out a compendium of ways in which group fairness definitions are unsatisfying. Yet despite these weaknesses, group fairness definitions are by far the most prevalent in the literature (see e.g. **???????** and **?** for a survey). This is in large part because notions of individual fairness require making stronger assumptions on the setting under consideration. In particular, the definition from **?** requires that the algorithm designer know a "task-specific fairness metric."

Learning problems over individuals are also often implicitly accompanied by some notion of *merit*, embedded in the objective function of the learning problem. For example, in a lending setting we might posit that each loan applicant is either "creditworthy" and will repay a loan, or is not creditworthy and will default — which is what we are trying to predict. **?** take the approach that this measure of merit — already present in the model, although initially unknown to the learner — can be taken to be the similarity metric in the definition of **?**, requiring informally that creditworthy individuals have at least the same probability of being accepted for loans as defaulting individuals. (The implicit and coarse fairness metric here assigns distance zero between pairs of creditworthy individuals and pairs of defaulting individuals, and some non-zero distance between a creditworthy

and a defaulting individual.) This resolves the problem of how one should discover the "fairness metric", but results in a notion of fairness that is necessarily aligned with the notion of "merit" (creditworthiness) that we are trying to predict.

However, there are many settings in which the notion of merit we wish to predict may be different or even at odds with the notion of fairness we would like to enforce. For example, notions of fairness aimed at rectifying societal inequities that result from historical discrimination can aim to favor the disadvantaged population (say, in college admissions), even if the performance of the admitted members of that population can be expected to be lower than that of the advantaged population. Similarly, we might desire a fairness metric incorporating only those attributes that individuals can change in principle (and thus excluding ones like race, age and gender), and that further expresses what are and are not meaningful differences between individuals, outside the context of any particular prediction problem. These kinds of fairness desiderata can still be expressed as an instantiation of the definition from **?**, but with a task-specific fairness metric separate from the notion of merit we are trying to predict.

In this paper, we revisit the individual fairness definition from **?**. This definition requires that pairs of individuals who are close in the fairness metric must be treated "similarly" (e.g. in an allocation problem such as lending, served with similar probability). We investigate the extent to which it is possible to satisfy this fairness constraint while simultaneously solving an online learning problem, when the underlying fairness metric is Mahalanobis but *not* known to the learning algorithm, and may also be in tension with the learning problem. One conceptual problem with metric-based definitions, that we seek to address, is that it may be difficult for anyone to actually precisely express a quantitative metric over individuals — but they nevertheless might "know unfairness when they see it." We therefore assume that the algorithm has access to an oracle that knows intuitively what it means to be fair, but cannot explicitly enunciate the fairness metric. Instead, given observed actions, the oracle can specify whether they were fair or not, and the goal is to obtain low regret in the online learning problem — measured with respect to the best *fair* policy — while also limiting violations of individual fairness during the learning process.

## 1.1   Our Results and Techniques

We study the standard linear contextual bandit setting. In rounds $t = 1, \ldots, T$, a learner observes arbitrary and possibly adversarially selected $d$-dimensional contexts, each corresponding to one of $k$ actions. The reward for each action is (in expectation) an unknown linear function of the contexts. The learner seeks to minimize its regret.

The learner also wishes to satisfy *fairness constraints*, defined with respect to an unknown distance function defined over contexts. The constraint requires that the difference between the probabilities that any two actions are taken is bounded by the distance between their contexts. The learner has no initial knowledge of the distance function. Instead, after the learner makes its decisions according to some probability distribution $\pi^t$ at round $t$, it receives feedback specifying for which pairs of contexts the fairness constraint was violated. Our goal in designing a learner is to simultaneously guarantee near-optimal regret in the contextual bandit problem (with respect to the best *fair* policy), while violating the fairness constraints as infrequently as possible. Our main result is a computationally efficient algorithm that guarantees this for a large class of distance functions known as *Mahalanobis distances* (these can be expressed as $d(x_1, x_2) = ||Ax_1 - Ax_2||_2$ for some matrix $A$).

**Theorem** (Informal): There is a computationally efficient learning algorithm $\boldsymbol{L}$ in our setting that guarantees that for any Mahalanobis distance, any time horizon $T$, and any error tolerance $\epsilon$:

1. (Learning) With high probability, $\boldsymbol{L}$ obtains regret $\tilde{O}\left(k^2 d^2 \log(T) + d\sqrt{T}\right)$ to the best fair policy (See Theorem 3 for a precise statement.)

2. (Fairness) With probability 1, $\boldsymbol{L}$ violates the unknown fairness constraints by more than $\epsilon$ on at most $O\left(k^2 d^2 \log(d/\epsilon)\right)$ many rounds. (Theorem 4.)

We note that the quoted regret bound requires setting $\epsilon = O(1/T)$, and so this implies a number of fairness violations of magnitude more than $1/T$ that is bounded by a function growing logarithmically in $T$. Other tradeoffs between regret and fairness violations are possible.

These two goals: of obtaining low regret, and violating the unknown constraint a small number of times — are seemingly in tension. A standard technique for obtaining a mistake bound with respect to fairness violations would be to play a "halving algorithm", which would always act as if the unknown metric is at the center of the current version space (the set of metrics consistent with the feedback observed thus far) — so that mistakes necessarily remove a non-trivial fraction of the version space, making progress. On the other hand, a standard technique for obtaining a diminishing regret bound is to play "optimistically" – i.e. to act as if the unknown metric is the point in the version space that would allow for the largest possible reward. But "optimistic" points are necessarily at the boundary of the version space, and when they are falsified, the corresponding mistakes do not necessarily reduce the version space by a constant fraction.

We prove our theorem in two steps. First, in Section 3, we consider the simpler problem in which the linear objective of the contextual bandit problem is known, and the distance function is all that is unknown. In this simpler case, we show how to obtain a bound on the number of fairness violations using a linear-programming based reduction to a recent algorithm which has a mistake bound for learning a linear function with a particularly weak form of feedback **?**. A complication is that our algorithm does not receive all of the feedback that the algorithm of **?** expects. We need to use the structure of our linear program to argue that this is ok. Then, in Section 4, we give our algorithm for the complete problem, using large portions of the machinery we develop in Section 3.

We note that in a non-adversarial setting, in which contexts are drawn from a distribution, the algorithm of **?** could be more simply applied along with standard techniques for contextual bandit learning to give an explore-then-exploit style algorithm. This algorithm would obtain bounded (but suboptimal) regret, and a number of fairness violations that grows as a root of $T$. The principal advantages of our approach are that we are able to give a number of fairness violations that has only *logarithmic* dependence on $T$, while tolerating contexts that are chosen adversarially, all while obtaining an optimal $O(\sqrt{T})$ regret bound to the best fair policy.

## 1.2 Additional Related Work

There are two papers, written concurrently to ours, that tackle orthogonal issues in metric-fair learning. **?** consider the problem of *generalization* when performing learning subject to a known metric constraint. They show that it is possible to prove relaxed PAC-style generalization bounds without any assumptions on the metric, and that for worst-case metrics, learning subject to a metric constraint can be computationally hard, even when the unconstrained learning problem is easy. In contrast, our work focuses on online learning with an *unknown* metric constraint. Our results imply similar generalization properties via standard online-to-offline reductions, but only for the class of metrics we study. **?** considers a group-fairness like relaxation of metric-fairness, asking that on average, individuals in pre-specified groups are classified with probabilities proportional to the average distance between individuals in those groups. They show how to learn such classifiers in the offline setting, given access to an oracle which can evaluate the distance between two individuals according to the metric (allowing for unbiased noise). The similarity to our work is that we also consider access to the fairness metric via an oracle, but our oracle is substantially weaker, and does not provide numeric valued output.

There are also several papers in the algorithmic fairness literature that are thematically related to ours, in that they both aim to bridge the gap between group notions of fairness (which can be semantically unsatisfying) and individual notions of fairness (which require very strong assumptions). **?** attempt to automatically learn a representation for the data in a batch learning problem (and hence, implicitly, a similarity metric) that causes a classifier to label an equal proportion of two protected groups as positive. They provide a heuristic approach and an experimental evaluation. Two recent papers (**?** and **?**) take the approach of asking for a group notion of fairness, but over exponentially many implicitly defined protected groups, thus mitigating what **?** call the "fairness gerrymandering" problem, which is one of the principal weaknesses of group fairness definitions. Both papers give polynomial time reductions which yield efficient algorithms whenever a corresponding agnostic learning problem is solvable. In contrast, in this paper, we take a different approach: we attempt to directly satisfy the original definition of individual fairness from **?**, but with substantially less information about the underlying similarity metric.

Starting with **?**, several papers have studied notions of fairness in classic and contextual bandit problems. **?** study a notion of "meritocratic" fairness in the contextual bandit setting, and prove upper

and lower bounds on the regret achievable by algorithms that must be "fair" at every round. This can be viewed as a variant of the **?** notion of fairness, in which the expected reward of each action is used to define the "fairness metric". The algorithm does not originally know this metric, but must discover it through experimentation. **?** extend the work of **?** to the setting in which the algorithm is faced with a continuum of options at each time step, and give improved bounds for the *linear* contextual bandit case. **?** extend this line of work to the reinforcement learning setting in which the actions of the algorithm can impact its environment. **?** consider a notion of fairness based on calibration in the simple stochastic bandit setting. Finally, **?** consider a notion of online group fairness in the stochastic contextual bandit setting by constraining how much probability mass can be placed on each pre-specified group of arms.

There is a large literature that focuses on learning Mahalanobis distances — see **?** for a survey. In this literature, the closest paper to our work focuses on *online* learning of Mahalanobis distances (**?**). However, this result is in a very different setting from the one we consider here. In **?**, the algorithm is repeatedly given pairs of points, and needs to predict their distance. It then learns their true distance, and aims to minimize its squared loss. In contrast, in our paper, the main objective of the learning algorithm is orthogonal to the metric learning problem — i.e. to minimize regret in the linear contextual bandit problem, but while simultaneously learning and obeying a fairness constraint, and only from weak feedback noting violations of fairness.

## 2 Model and Preliminaries

### 2.1 Linear Contextual Bandits

We study algorithms that operate in the *linear contextual bandits* setting. A linear contextual bandit problem is parameterized by an unknown vector of linear coefficients $\theta \in \mathbb{R}^d$, with $||\theta||_2 \leq 1$. Algorithms in this setting operate in *rounds* $t = 1, \ldots, T$. In each round $t$, an algorithm $\boldsymbol{L}$ observes $k$ *contexts* $x_1^t, \ldots, x_k^t \in \mathbb{R}^d$, scaled such that $||x_i^t||_2 \leq 1$. We write $x^t = (x_1^t, \ldots, x_k^t)$ to denote the entire set of contexts observed at round $t$. After observing the contexts, the algorithm chooses an action $i^t$. After choosing an action, the algorithm obtains some stochastic *reward* $r_{i^t}^t$ such that $r_{i^t}^t$ is subgaussian[1] and $\mathbb{E}[r_{i^t}^t] = \langle x_{i^t}^t, \theta \rangle$. The algorithm does not observe the reward for the actions not chosen. When the action $i^t$ is clear from context, and write $r^t$ instead of $r_{i^t}^t$.

**Remark 1.** *For simplicity, we consider algorithms that select only a* single *action at every round. However, this assumption is not necessary. In the appendix of the full version (**?**), we show how our results extend to the case in which the algorithm can choose any number of actions at each round. This assumption is sometimes more natural: for example, in a lending scenario, a bank may wish to make loans to as many individuals as will be profitable, without a budget constraint.*

In this section, we will be discussing algorithms $\boldsymbol{L}$ that are necessarily randomized. To formalize this, we denote a history including everything observed by the algorithm up through but not including round $t$ as $h^t = ((x^1, i^1, r^1), \ldots, (x^{t-1}, i^{t-1}, r^{t-1}))$ The space of such histories is denoted by $\mathcal{H}^t = (\mathbb{R}^{d \times k} \times [k] \times \mathbb{R})^{t-1}$. An algorithm $\boldsymbol{L}$ is defined by a sequence of functions $f^1, \ldots, f^T$ each mapping histories and observed contexts to probability distributions over actions: $f^t : \mathcal{H}^t \times \mathbb{R}^{d \times k} \to \Delta[k]$. We write $\pi^t$ to denote the probability distribution over actions that $\boldsymbol{L}$ plays at round $t$: $\pi^t = f^t(h^t, x^t)$. We view $\pi^t$ as a vector over $[0,1]^k$, and so $\pi_i^t$ denotes the probability that $\boldsymbol{L}$ plays action $i$ at round $t$. We denote the expected reward of the algorithm at day $t$ as $\mathbb{E}[r^t] = \mathbb{E}_{i \sim \pi^t}[r_i^t]$. It will sometimes also be useful to refer to the vector of expected rewards across all actions on day $t$. We denote it as $\bar{r}^t = (\langle x_1^t, \theta \rangle, \ldots, \langle x_k^t, \theta \rangle)$.

### 2.2 Fairness Constraints and Feedback

We study algorithms that are constrained to behave *fairly* in some manner. We adapt the definition of fairness from **?** that asserts, informally, that "similar individuals should be treated similarly". We imagine that the decisions that our contextual bandit algorithm $\boldsymbol{L}$ makes correspond to individuals, and that the contexts $x_i^t$ correspond to features pertaining to individuals. We adopt the following (specialization of) the fairness definition from Dwork et al, which is parameterized by a distance function $d : \mathbb{R}^d \times \mathbb{R}^d \to \mathbb{R}$.

**Definition 1** (**?**). *Algorithm $\boldsymbol{L}$ is Lipschitz-fair on round $t$ with respect to distance function $d$ if for all pairs of individuals $i, j$: $|\pi_i^t - \pi_j^t| \leq d(x_i^t, x_j^t)$. For brevity, we will often just say that the algorithm is* fair *at round $t$, with the understanding that we are always talking about this one particular kind of fairness.*

**Remark 2.** *Note that this definition requires a fairness constraint that binds between individuals at a single round $t$, but not between rounds $t$. This is for several reasons. First, at a philosophical level, we want our algorithms to be able to improve with time, without being bound by choices they made long ago before they had any information about the fairness metric. At a (related) technical level, it is easy to construct lower bound instances that certify that it is impossible to simultaneously guarantee that an algorithm has diminishing regret to the best fair policy, while violating fairness constraints (now defined as binding across rounds) a sublinear number of times.*

One of the main difficulties in working with Lipschitz fairness (as discussed in **?**) is that the distance function $d$ plays a central role, but it is not clear how it should be specified. In this paper, we concern ourselves with learning $d$ from feedback. In particular, algorithms $\boldsymbol{L}$ will have access to a *fairness oracle*, which models a regulator who "knows unfairness when he sees it".

Informally, the fairness oracle will take as input: 1) the set of choices available to $\boldsymbol{L}$ at each round $t$, and 2) the probability distribution $\pi^t$ that $\boldsymbol{L}$ uses to make its choices at round $t$, and returns the set of all pairs of individuals for which $\boldsymbol{L}$ violates the fairness constraint.

**Definition 2** (Fairness Oracle). *Given a distance function $d$, a fairness oracle $\boldsymbol{O}_d$ is a function $\boldsymbol{O}_d : \mathbb{R}^{d \times k} \times \Delta[k] \to 2^{[k] \times [k]}$ defined such that: $\boldsymbol{O}_d(x^t, \pi^t) = \{(i, j) : |\pi_i^t - \pi_j^t| > d(x_i^t, x_j^t)\}$*

Formally, algorithms $\boldsymbol{L}$ in our setting will operate in the following environment:

1. An adversary fixes a linear reward function $\theta \in \mathbb{R}^d$ with $||\theta|| \leq 1$ and a distance function $d$. $\boldsymbol{L}$ is given access to the fairness oracle $\boldsymbol{O}_d$.

2. In rounds $t = 1$ to $T$:

   (a) The adversary chooses contexts $x^t \in \mathbb{R}^{d \times k}$ with $||x_i^t|| \leq 1$ and gives them to $\boldsymbol{L}$.

   (b) $\boldsymbol{L}$ chooses a probability distribution $\pi^t$ over actions, and chooses action $i^t \sim \pi^t$.

   (c) $\boldsymbol{L}$ receives reward $r_{i^t}^t$ and observes feedback $\boldsymbol{O}_d(\pi^t)$ from the fairness oracle.

Because of the power of the adversary in this setting, it is not possible to avoid arbitrarily small violations of the fairness constraint. Instead, we will aim to limit *significant* violations.

**Definition 3.** *Algorithm $\boldsymbol{L}$ is $\epsilon$-unfair on pair $(i, j)$ at round $t$ with respect to distance function $d$ if $|\pi_i^t - \pi_j^t| > d(x_i^t, x_j^t) + \epsilon$. Given a sequence of contexts and a history $h^t$ (which fixes the distribution on actions at day $t$) We write $\mathbf{Unfair}(\boldsymbol{L}, \epsilon, h^t) = \sum_{i=1}^{k-1} \sum_{j=i+1}^{k} \mathbb{1}(|\pi_i^t - \pi_j^t| > d(x_i^t, x_j^t) + \epsilon)$ to denote the number of pairs on which $\boldsymbol{L}$ is $\epsilon$-unfair at round $t$.*

Given a distance function $d$ and a history $h^{T+1}$, the *$\epsilon$-fairness loss* of an algorithm $\boldsymbol{L}$ is the total number of pairs on which it is $\epsilon$-unfair: $\mathbf{FairnessLoss}(\boldsymbol{L}, h^{T+1}, \epsilon) = \sum_{t=1}^{T} \mathbf{Unfair}(\boldsymbol{L}, \epsilon, h^t)$ For a shorthand, we write $\mathbf{FairnessLoss}(\boldsymbol{L}, T, \epsilon)$.

We will aim to design algorithms $\boldsymbol{L}$ that guarantee that their fairness loss is bounded with probability 1 in the worst case over the instance: i.e. in the worst case over both $\theta$ and $x^1, \ldots, x^T$, and in the worst case over the distance function $d$ (within some allowable class of distance functions – see Section 2.4).

## 2.3 Regret to the Best Fair Policy

In addition to minimizing fairness loss, we wish to design algorithms that exhibit diminishing *regret* to the best *fair* policy. We first define a linear program that we will make use of throughout the paper. Given a vector $a \in \mathbb{R}^d$ and a vector $c \in \mathbb{R}^{k^2}$, we denote by $LP(a, c)$ the following linear program:

$$
\begin{aligned}
\underset{\pi=\{p_1,\dots,p_k\}}{\text{maximize}} \quad & \sum_{i=1}^{k} p_i a_i \\
\text{subject to} \quad & |p_i - p_j| \le c_{i,j}, \forall (i,j) \\
& \sum_{i=1}^{k} p_i \le 1
\end{aligned}
$$

We write $\pi(a,c) \in \Delta[k]$ to denote an optimal solution to $LP(a,c)$. Given a set of contexts $x^t$, recall that $\bar{r}^t$ is the vector representing the expected reward corresponding to each context (according to the true, unknown linear reward function $\theta$). Similarly, we write $\bar{d}^t$ to denote the vector representing the set of distances between each pair of contexts $i,j$ (according to the true, unknown distance function $d$): $\bar{d}^t_{i,j} = d(x_i^t, x_j^t)$.

Observe that $\pi(\bar{r}^t, \bar{d}^t)$ corresponds to the distribution over actions that maximizes expected reward at round $t$, subject to satisfying the fairness constraints — i.e. the distribution that an optimal player, with advance knowledge of $\theta$ would play, if he were not allowed to violate the fairness constraints at all. This is the benchmark with respect to which we define regret:

**Definition 4.** *Given an algorithm $\boldsymbol{L}$ ($f_1, \dots, f_T$), a distance function $d$, a linear parameter vector $\theta$, and a history $h^{T+1}$ (which includes a set of contexts $x^1, \dots, x^T$), its regret is defined to be:*

$$
\textbf{Regret}(\boldsymbol{L}, \theta, d, h^{T+1}) = \sum_{t=1}^{T} \underset{i \sim \pi(\bar{r}^t, \bar{d}^t)}{\mathbb{E}} [\bar{r}_i^t] - \sum_{t=1}^{T} \underset{i \sim f^t(h^t, x^t)}{\mathbb{E}} [\bar{r}_i^t]
$$

Our goal will be to design algorithms for which we can bound regret with high probability over the randomness of $h^{T+1}$ in the worst case over $\theta$, $d$, and $(x^1, \dots, x^T)$.

### 2.4 Mahalanobis Distance

In this paper, we will restrict our attention to a special family of distance functions which are parameterized by a matrix $A$:

**Definition 5** (Mahalanobis distances). *A function $d : \mathbb{R}^d \times \mathbb{R}^d \to \mathbb{R}$ is a Mahalanobis distance function if there exists a matrix $A$ such that for all $x_1, x_2 \in \mathbb{R}^d$: $d(x_1, x_2) = ||Ax_1 - Ax_2||_2$ where $||\cdot||_2$ denotes Euclidean distance. Note that if $A$ is not full rank, then this does not define a metric — but we will allow this case (and be able to handle it in our algorithmic results).*

Mahalanobis distances will be convenient for us to work with, because *squared* Mahalanobis distances can be expressed as follows:

$$
\begin{aligned}
d(x_1, x_2)^2 &= ||Ax_1 - Ax_2||_2^2 = \langle A(x_1 - x_2), A(x_1 - x_2) \rangle \\
&= (x_1 - x_2)^\top A^\top A(x_1 - x_2) = \sum_{i,j=1}^{d} G_{i,j}(x_1 - x_2)_i (x_1 - x_2)_j
\end{aligned}
$$

where $G = A^\top A$. Observe that when $x_1$ and $x_2$ are fixed, this is a linear function in the entries of the matrix $G$. We will use this property to reason about *learning $G$*, and thereby learning $d$.

## 3 Warmup: The Known Objective Case

In this section, we consider an easier case of the problem in which the linear objective function $\theta$ is known to the algorithm, and the distance function $d$ is all that is unknown. In this case, we show via a reduction to an online learning algorithm of **?**, how to simultaneously obtain a logarithmic regret bound and a logarithmic (in $T$) number of fairness violations. The analysis we do here will be useful when we solve the full version of our problem (in which $\theta$ is unknown) in Section 4. Here, we sketch our solution. Details are in the full version of the paper (**?**).

## 3.1 Outline of the Solution

Recall that since we know $\theta$, at every round $t$ after seeing the contexts, we know the vector of expected rewards $\bar{r}^t$ that we would obtain for selecting each action. Our algorithm will play at each round $t$ the distribution $\pi(\bar{r}^t, \hat{d}^t)$ that results from solving the linear program $LP(\bar{r}^t, \hat{d}^t)$, where $\hat{d}^t$ is a "guess" for the pairwise distances between each context $\bar{d}^t$. (Recall that the optimal distribution to play at each round is $\pi(\bar{r}^t, \bar{d}^t)$.)

The main engine of our reduction is an efficient online learning algorithm for linear functions recently given by **?**. Their algorithm, which we refer to as **DistanceEstimator**, works in the following setting. There is an unknown vector of linear parameters $\alpha \in \mathbb{R}^m$. In rounds $t$, the algorithm observes a vector of features $u^t \in \mathbb{R}^m$, and produces a prediction $g^t \in \mathbb{R}$ for the value $\langle \alpha, u^t \rangle$. After it makes its prediction, the algorithm learns whether its guess was *too large* or not, but does not learn anything else about the value of $\langle \alpha, u^t \rangle$. The guarantee of the algorithm is that the number of rounds in which its prediction is off by more than $\epsilon$ is bounded by $O(m \log(m/\epsilon))^2$.

Our strategy will be to instantiate $\binom{k}{2}$ copies of this distance estimator — one for each pair of actions — to produce guesses $(\hat{d}_{i,j}^t)^2$ intended to approximate the *squared* pairwise distances $d(x_i^t, x_j^t)^2$. From this we derive estimates $\hat{d}_{i,j}^t$ of the pairwise distances $d(x_i^t, x_j^t)$. Note that this is a linear estimation problem for any Mahalanobis distance, because by our observation in Section 2.4, a squared Mahalanobis distance can be written as a linear function of the $m = d^2$ unknown entries of the matrix $G = A^\top A$ which defines the Mahalanobis distance.

The complication is that the **DistanceEstimator** algorithms expect feedback at every round, which we cannot always provide. This is because the fairness oracle $O_d$ provides feedback about the distribution $\pi(\bar{r}^t, \hat{d}^t)$ used by the algorithm, *not* directly about the guesses $\hat{d}^t$. These are not the same, because not all of the constraints in the linear program $LP(\bar{r}^t, \hat{d}^t)$ are necessarily tight — it may be that $|\pi(\bar{r}^t, \hat{d}^t)_i - \pi(\bar{r}^t, \hat{d}^t)_j| < \hat{d}_{i,j}^t$. For any copy of **DistanceEstimator** that does not receive feedback, we can simply "roll back" its state and continue to the next round. But we need to argue that we make progress — that whenever we are $\epsilon$-unfair, or whenever we experience large per-round regret, then there is at least one copy of **DistanceEstimator** that we can give feedback to such that the corresponding copy of **DistanceEstimator** has made a large prediction error, and we can thus charge either our fairness loss or our regret to the mistake bound of that copy of **DistanceEstimator**.

As we show, there are three relevant cases.

1. In any round in which we are $\epsilon$-unfair for some pair of contexts $x_i^t$ and $x_j^t$, then it must be that $\hat{d}_{i,j}^t \geq d(x_i^t, x_j^t) + \epsilon$, and so we can always update the $(i,j)$th copy of **DistanceEstimator** and charge our fairness loss to its mistake bound.

2. For any pair of arms $(i,j)$ such that we have not violated the fairness constraint, *and* the $(i,j)$th constraint in the linear program is tight, we can provide feedback to the $(i,j)$th copy of **DistanceEstimator** (its guess was not too large). There are two cases. Although the algorithm never knows which case it is in, we handle each case separately in the analysis.

   (a) For every constraint $(i,j)$ in $LP(\bar{r}^t, \hat{d}^t)$ that is *tight* in the optimal solution, $|\hat{d}_{i,j}^t - d(x_i^t, x_j^t)| \leq \epsilon$. In this case, we show that our algorithm does not incur very much per round regret.

   (b) Otherwise, there is a tight constraint $(i,j)$ such that $|\hat{d}_{i,j}^t - d(x_i^t, x_j^t)| > \epsilon$. In this case, we may incur high per-round regret — but we can charge such rounds to the mistake bound of the $(i,j)$th copy of **DistanceEstimator**.

**Theorem 1.** $\textbf{FairnessLoss}(\boldsymbol{L}_{known-\theta}, T, \epsilon) \leq O\left(k^2 d^2 \log\left(\frac{d \cdot ||A^T A||_F}{\epsilon}\right)\right)$

**Theorem 2.** *For any time horizon $T$:* $\textbf{Regret}(\boldsymbol{L}_{known-\theta}, T) \leq O\left(k^2 d^2 \log\left(\frac{d \cdot ||A^\top A||_F}{\epsilon}\right) + k^3 \epsilon T\right)$

*Setting $\epsilon = O(1/(k^3 T))$ yields a regret bound of $O(d^2 \log(||A^\top A||_F \cdot dkT))$.*

## 4 The Full Algorithm

### 4.1 Outline of the Solution

At a high level, our plan will be to combine the techniques used for the case where the linear objective $\theta$ is known with a standard "optimism in the face of uncertainty" strategy for learning the parameter vector $\theta$. Our algorithm will maintain a ridge-regression estimate $\tilde{\theta}$ together with confidence regions derived in **?**. After it observes the contexts $x_i^t$ at round $t$, it uses these to derive upper confidence bounds on the expected rewards, corresponding to each context — represented as a vector $\hat{r}^t$. The algorithm continues to maintain distance estimates, $\hat{d}^t$, the same way as the case where the linear objective $\theta$ is known, using **?** as a subroutine. At ever round, the algorithm then chooses its action according to the distribution $\pi^t = \pi(\hat{r}^t, \hat{d}^t)$.

The regret analysis of the algorithm follows by decomposing the per-round regret into two pieces. The first can be bounded by the sum of the *expected widths* of the confidence intervals corresponding to each context $x_i^t$ that might be chosen at each round $t$, where the expectation is over the randomness of the algorithm's distribution $\pi^t$. A theorem of **?** bounds the sum of the widths of the confidence intervals corresponding to arms *actually chosen* by the algorithm. Using a martingale concentration inequality, we are able to relate these two quantities. We show that the second piece of the regret bound can be manipulated into a form that can be bounded using machinery built in section 3, which is described in further details in the full version (**?**).

**Theorem 3.** *For any time horizon $T$, with probability $1 - \delta$:*

$$\mathbf{Regret}(\boldsymbol{L}_{full}, T) \leq O\left(k^2 d^2 \log\left(\frac{d \cdot ||A^\top A||_F}{\epsilon}\right) + k^3 \epsilon T + d\sqrt{T}\log(\frac{T}{\delta})\right)$$

*If $\epsilon = 1/k^3 T$, this is a regret bound of $O\left(k^2 d^2 \log\left(kdT \cdot ||A^\top A||_F\right) + d\sqrt{T}\log(\frac{T}{\delta})\right)$*

Finally, the bound on the fairness loss is identical to the bound we proved in Theorem 1 (because our algorithm for constructing distance estimates $\hat{d}$ is unchanged). We have:

**Theorem 4.** *For any sequence of contexts and any Mahalanobis distance $d(x_1, x_2) = ||Ax_1 - Ax_2||_2$:*

$$\mathbf{FairnessLoss}(\boldsymbol{L}_{full}, T, \epsilon) \leq O\left(k^2 d^2 \log\left(\frac{d \cdot ||A^\top A||_F}{\epsilon}\right)\right)$$

## 5 Conclusion and Future Directions

We have initiated the study of fair sequential decision making in settings where the notions of payoff and fairness are separate and may be in tension with each other, and have shown that in a stylized setting, optimal fair decisions can be efficiently learned *even without direct knowledge of the fairness metric*. A number of extensions of our framework and results would be interesting to examine. At a high level, the interesting question is: how much can we further relax the information about the fairness metric available to the algorithm? For instance, what if the fairness feedback is only partial, identifying some but not all fairness violations? What if it only indicates whether or not there were any violations, but does not identify them? What if the feedback is not guaranteed to be exactly consistent with any metric? Or what if the feedback is consistent with *some* distance function, but not one in a known class: for example, what if the distance is not exactly Mahalanobis, but is approximately so? In general, it is very interesting to continue to push to close the wide gap between the study of individual fairness notions and the study of group fairness notions. When can we obtain the strong semantics of individual fairness without making correspondingly strong assumptions?

## Footnotes

[1]A random variable $X$ with $\mu = \mathbb{E}[X]$ is sub-gaussian, if for all $t \in \mathbb{R}$, $\mathbb{E}[e^{t(X-\mu)}] \leq e^{\frac{t^2}{2}}$.

[2]If the algorithm also learned whether or not its guess was in error by more than $\epsilon$ at each round, variants of the classical halving algorithm could obtain this guarantee. But the algorithm does not receive this feedback, which is why the more sophisticated algorithm of **?** is needed.

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
