[Supplementary Material]

# Online Learning with an Unknown Fairness Metric

Stephen Gillen[*]     Christopher Jung[†]     Michael Kearns[‡]     Aaron Roth[§]

September 17, 2018

## Abstract

We consider the problem of online learning in the linear contextual bandits setting, but in which there are also strong *individual fairness* constraints governed by an unknown similarity metric. These constraints demand that we select similar actions or individuals with approximately equal probability [Dwork et al., 2012], which may be at odds with optimizing reward, thus modeling settings where profit and social policy are in tension. We assume we learn about an unknown Mahalanobis similarity metric from only weak feedback that identifies fairness violations, but does not quantify their extent. This is intended to represent the interventions of a regulator who "knows unfairness when he sees it" but nevertheless cannot enunciate a quantitative fairness metric over individuals. Our main result is an algorithm in the adversarial context setting that has a number of fairness violations that depends only logarithmically on $T$, while obtaining an optimal $O(\sqrt{T})$ regret bound to the best fair policy.

---

[*]Department of Mathematics, University of Pennsylvania.

[†]Department of Computer and Information Sciences, University of Pennsylvania. Supported in part by a grant from the Quattrone Center for the Fair Administration of Justice.

[‡]Department of Computer and Information Sciences, University of Pennsylvania.

[§]Department of Computer and Information Sciences, University of Pennsylvania. Supported in part by grants from the DARPA Brandeis project, the Sloan Foundation, and NSF grants CNS-1513694 and CNS-1253345.

# 1 Introduction

The last several years have seen an explosion of work studying the problem of fairness in machine learning. Yet there remains little agreement about what "fairness" should mean in different contexts. In broad strokes, the literature can be divided into two families of fairness definitions: those aiming at *group* fairness, and those aiming at *individual* fairness.

Group fairness definitions are aggegate in nature: they partition individuals into some collection of *protected groups* (say by race or gender), specify some statistic of interest (say, positive classification rate or false positive rate), and then require that a learning algorithm equalize this quantity across the protected groups. On the other hand, individual fairness definitions ask for some constraint that binds on the individual level, rather than only over averages of people. Often, these constraints have the semantics that "similar people should be treated similarly" Dwork et al. [2012].

Individual fairness definitions have substantially stronger semantics and demands than group definitions of fairness. For example, Dwork et al. [2012] lay out a compendium of ways in which group fairness definitions are unsatisfying. Yet despite these weaknesses, group fairness definitions are by far the most prevalent in the literature (see e.g. Kamiran and Calders [2012], Hajian and Domingo-Ferrer [2013], Kleinberg et al. [2017], Hardt et al. [2016], Friedler et al. [2016], Zafar et al. [2017], Chouldechova [2017] and Berk et al. [2017]

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

## 3.2   The Distance Estimator

First, we fix some notation for the **DistanceEstimator** algorithm. We write **DistanceEstimator**$(\epsilon)$ to instantiate a copy of **DistanceEstimator** with a mistake bound for $\epsilon$-misestimations. The mistake bound we state for **DistanceEstimator** is predicated on the assumption that the norm of the unknown linear parameter vector $\alpha \in \mathbb{R}^m$ is bounded by $\|\alpha\| \leq B_1$, and the norms of the arriving vectors $u^t \in \mathbb{R}^m$ are bounded by $\|u^t\| \leq B_2$. Given an instantiation of **DistanceEstimator** and a new vector $u^t$ for which we would like a prediction, we write: $g^t = $ **DistanceEstimator**$.guess(u^t)$ for its guess of the value of $\langle \alpha, u^t \rangle$. We use the following notation to refer to the feedback we provide to **DistanceEstimator**: If $g^t > \langle \alpha, u^t \rangle$ and we provide feedback, we write **DistanceEstimator**$.feedback(\top)$. Otherwise, if $g^t \leq \langle \alpha, u^t \rangle$ and we give feedback, we write **DistanceEstimator**$.feedback(\bot)$. In some rounds, we may be unable to provide the feedback that **DistanceEstimator** is expecting: in these rounds, we simply "roll-back" its internal state. We can do this because the mistake bound for **DistanceEstimator** holds for *every* sequence of arriving vectors $u^t$. If we give feedback to **DistanceEstimator** in a given round $t$, we write $v^t = 1$ write $v^t = 0$ otherwise.

**Definition 7.** *Given an accuracy parameter $\epsilon$, a linear parameter vector $\alpha$, a sequence of vectors $u^1, \ldots, u^T$, a sequence of guesses $g^1, \ldots, g^T$ and a sequence of feedback indicators, $v^1, \ldots, v^T$, the number of valid $\epsilon$-mistakes made by* **DistanceEstimator** *is:*

$$\mathbf{Mistakes}(\epsilon) = \sum_{t=1}^{T} \mathbb{1}(v^t = 1 \wedge |g^t - \langle u^t, \alpha \rangle| > \epsilon)$$

*In other words, it is the number of $\epsilon$-mistakes made by* **DistanceEstimator** *in rounds for which we provided the algorithm feedback.*

We now state a version of the main theorem from Lobel et al. [2017], adapted to our setting[4]:

**Lemma 1** (Lobel et al. [2017]). *For any $\epsilon > 0$ and any sequence of vectors $u^1, \ldots, u^T$,* **DistanceEstimator**$(\epsilon)$ *makes a bounded number of valid $\epsilon$-mistakes.*

$$\mathbf{Mistakes}(\epsilon) = O\left( m \log\left( \frac{m \cdot B_1 \cdot B_2}{\epsilon} \right) \right)$$

## 3.3   The Algorithm

For each pair of arms $i, j \in [k]$, our algorithm instantiates a copy of **DistanceEstimator**$(\epsilon^2)$, which we denote by **DistanceEstimator**$_{i,j}$: we also subscript all variables relevant to **DistanceEstimator**$_{i,j}$ with $i, j$ (e.g. $u_{i,j}^t$). The underlying linear parameter vector we want to learn $\alpha = flatten(G) \in \mathbb{R}^{d^2}$, where $flatten : \mathbb{R}^{m \times n} \to \mathbb{R}^{m \cdot n}$ maps a $m \times n$ matrix to a vector of size $mn$ by concatenating its rows into a vector. Similarly, given a pair of contexts $x_i^t, x_j^t$, we will define $u_{i,j}^t = flatten((x_i^t - x_j^t)(x_i^t - x_j^t)^\top)$. **DistanceEstimator**$_{i,j}.guess(u_{i,j}^t)$ will output guess $g_{i,j}^t$ for the value $\langle \alpha, u_{i,j}^t \rangle = (\bar{d}_{i,j}^t)^2$, as

$$\langle flatten(G), flatten((x_i^t - x_j^t)(x_i^t - x_j^t)^\top) \rangle = \sum_{a,b=1}^{d} G_{a,b}(x_i^t - x_j^t)_a (x_i^t - x_j^t)_b = (\bar{d}_{i,j}^t)^2$$

We take $\hat{d}_{i,j}^t = \sqrt{g_{i,j}^t}$ as our estimate for the distance between $x_i^t$ and $x_j^t$.

The algorithm then chooses an arm to pull according to the distribution $\pi(\bar{r}^t, \hat{d}^t)$, where $\bar{r}_i^t = \langle \theta, x_i \rangle$. The fairness oracle $O_d$ returns all pairs of arms that violate the fairness constraints. For these pairs $(i, j)$ we provide feedback to **DistanceEstimator**$_{i,j}$: the guess was too large. For the remaining pairs of arms $(i, j)$, there are two cases. If the $(i, j)$th constraint in $LP(\bar{r}^t, \hat{d}^t)$ was not tight, then we provide no feedback ($v_{i,j}^t = 0$). Otherwise, we provide feedback: the guess was not too large. The pseudocode appears as Algorithm 1.

First we derive the valid mistake bound that the **DistanceEstimator**$_{i,j}$ algorithms incur in our parameterization.

**Lemma 2.** *For pair $(i, j)$, the total number of valid $\epsilon^2$ mistakes made by* **DistanceEstimator**$_{i,j}$ *is bounded as:*

$$\mathbf{Mistakes}(\epsilon^2) = O\left( d^2 \log\left( \frac{d \cdot \|A^\top A\|_F}{\epsilon} \right) \right)$$

*where the distance function is defined as $d(x_i, x_j) = \|Ax_i - Ax_j\|_2$ and $\|\cdot\|_F$ denotes the Frobenius norm.*

**for** $i, j = 1, \ldots, k$ **do**
　　$\vert$　**DistanceEstimator**$_{i,j}$ = **DistanceEstimator**$(\epsilon^2)$
**end**
**for** $t = 1, \ldots, T$ **do**
　　receive the contexts $x^t = (x_1^t, \ldots, x_k^t)$
　　**for** $i, j = 1, \ldots, k$ **do**
　　　$\vert$　$u_{i,j}^t = flatten((x_i^t - x_j^t)(x_i^t - x_j^t)^\top)$
　　　$\vert$　$g_{i,j}^t = $ **DistanceEstimator**$_{ij}.guess(u_{i,j}^t)$
　　　$\vert$　$\hat{d}_{i,j}^t = \sqrt{g_{i,j}^t}$
　　**end**
　　$\pi^t = \pi(\bar{r}^t, \hat{d}^t)$
　　Pull an arm $i^t$ according to $\pi^t$ and receive a reward $r_{i^t}^t$
　　$S = O_d(x^t, \pi^t)$
　　$R = \{(i, j) | (i, j) \notin S \wedge |p_i^t - p_j^t| = \hat{d}_{ij}^t\}$
　　**for** $(i, j) \in S$ **do**
　　　$\vert$　**DistanceEstimator**$_{ij}.feedback(\bot)$
　　　$\vert$　$v_{ij}^t = 1$
　　**end**
　　**for** $(i, j) \in R$ **do**
　　　$\vert$　**DistanceEstimator**$_{ij}.feedback(\top)$
　　　$\vert$　$v_{ij}^t = 1$
　　**end**
**end**

**Algorithm 1:** $L_{\mathrm{known}-\theta}$

*Proof.* This follows directly from Lemma 1, and the observations that in our setting, $m = d^2$, $B_1 = \|\alpha\| = \|A^\top A\|_F$, and

$$B_2 \leq \max_t \|u_{i,j}^t\|_2 \leq \max_t \|(x_i^t - x_j^t)\|^2 \leq 4.$$

$\square$

We next observe that since we only instantiate $k^2$ copies of **DistanceEstimator** in total, Lemma 2 immediately implies the following bound on the total number of rounds in which *any* distance estimator that receives feedback provides us with a distance estimate that differs by more than $\epsilon$ from the correct value:

**Corollary 1.** *The number of rounds where there exists a pair $(i,j)$ such that feedback is provided $(v_{i,j}^t = 1)$ and its estimate is off by more than $\epsilon$ is bounded:*

$$\left|\{t : \exists (i,j) : v_{ij}^t = 1 \wedge |\hat{d}_{i,j}^t - \bar{d}_{i,j}^t| > \epsilon\}\right| \leq O\left(k^2 d^2 \log\left(\frac{d \cdot \|A^\top A\|_F}{\epsilon}\right)\right)$$

*Proof.* This follows from summing the $k^2$ valid $\epsilon^2$ mistake bounds for each copy of **DistanceEstimator**$_{i,j}$, and noting that an $\epsilon$ mistake in predicting the value of $\bar{d}_{i,j}^t$ implies an $\epsilon^2$ mistake in predicting the value of $(\bar{d}_{i,j}^t)^2$. $\square$

We now have the pieces to bound the $\epsilon$-unfairness loss of our algorithm:

**Theorem 1.** *For any sequence of contexts and any Mahalanobis distance $d(x_1, x_2) = \|Ax_1 - Ax_2\|_2$:*

$$\textbf{FairnessLoss}(L_{known-\theta}, T, \epsilon) \leq O\left(k^2 d^2 \log\left(\frac{d \cdot \|A^T A\|_F}{\epsilon}\right)\right)$$

*Proof.*

$$\textbf{FairnessLoss}(L_{\text{known}-\theta}, T, \epsilon) = \sum_{t=1}^T \textbf{Unfair}(L_{\text{known}-\theta}, \epsilon)$$

$$\leq \sum_{t=1}^T \sum_{i,j} \mathbb{1}(|\pi_i^t - \pi_j^t| > \bar{d}_{ij}^t + \epsilon)$$

$$= \sum_{i,j} \sum_{t=1}^T \mathbb{1}(\{v_{ij}^t = 1 \wedge \hat{d}_{ij}^t > d_{ij}^t + \epsilon\})$$

$$\leq \sum_{i,j} \sum_{t=1}^T \mathbb{1}(\{v_{ij}^t = 1 \wedge |\hat{d}_{ij}^t - d_{ij}^t| > \epsilon\})$$

$$= O\left(k^2 d^2 \log\left(\frac{d \cdot \|A^\top A\|_F}{\epsilon}\right)\right) \qquad \text{Corollary 1}$$

$\square$

Figure 1: A visual interpretation of the surgery performed on $p$ in the proof of Lemma 3 to obtain $P'$. Note that the surgery manages to shrink the distance between $p_a$ and $p_b$ without increasing the distance between any other pair of points.

We now turn our attention to bounding the regret of the algorithm. Recall from the overview in Section 3.1, that our plan will be to divide rounds into two types. In rounds of the first type, our distance estimates corresponding to every *tight constraint* in the linear program have only small error. We cannot bound the number of such rounds, but we can bound the regret incurred in any such rounds. In rounds of the second type, we have at least one significant error in the distance estimate corresponding to a tight constraint. We might incur significant regret in such rounds, but we can bound the number of such rounds.

The following lemma bounds the *decrease* in expected per-round reward that results from under-estimating a *single* distance constraint in our linear programming formulation.

**Lemma 3.** *Fix any vector of distance estimates $d$ and any vector of rewards $r$. Fix a constant $\epsilon$ and any pair of coordinates $(a,b) \in [k] \times [k]$. Let $d'$ be the vector such that $d'_{ab} = d_{ab} - \epsilon$ and $d'_{ij} = d_{ij}$ for*

$(i,j) \neq (a,b)$, *then* $\langle r, \pi(r,d) \rangle - \langle r, \pi(r,d') \rangle \leq \epsilon \sum_{i=1}^{k} r_i$

*Proof.* The plan of the proof is to start with $\pi(r,d)$ and perform surgery on it to arrive at a new probability distribution $p' \in \Delta k$ that satisfies the constraints of $LP(r,d')$, and obtains objective value at least $\langle r, p' \rangle \geq \langle r, \pi(r,d) \rangle - \epsilon \sum_{i=1}^{k} r_i$. Because $p'$ is feasible, it lower bounds the objective value of the optimal solution $\pi(r,d')$, which yields the theorem.

To reduce notational clutter, for the rest of the argument we write $p$ to denote $\pi(r,d)$. Without loss of generality, we assume that $p_a \geq p_b$. If $p_a - p_b \leq d_{ab} - \epsilon$, then $p_i$ is still a feasible solution to $LP(r,d')$, and we are done. Thus, for the rest of the argument, we can assume that $p_a - p_b > d_{ab} - \epsilon$. We write $\Delta = (p_a - p_b) - (d_{ab} - \epsilon) > 0$

We now define our modified distribution $p'$:

$$p'_i = \begin{cases} p_i - \Delta & p_a \leq p_i \\ p_a - \Delta & p_a - \Delta \leq p_i < p_a \\ p_i & \text{otherwise} \end{cases}$$

We'll partition the coordinates of $p_i$ into which of the three cases they fall into in our definition of $p'$ above. $S_1 = \{i | p_a \leq p_i\}$, $S_2 = \{i | p_a - \epsilon \leq p_i < p_a\}$, and $S_3 = \{i | i < p_b + (d_{ab} - \epsilon)\}$. It remains to verify that $p'$ is a feasible solution to $LP(r,d')$, and that it obtains the claimed objective value.

**Feasibility:** First, observe that $\sum_i p_i' \leq 1$. This follows because $p'$ is coordinate-wise smaller than $p$, and by assumption, $p$ was feasible. Thus, $\sum_i p_i' \leq \sum_i p_i \leq 1$.

Next, observe that by construction, $p_i' \geq 0$ for all $i$. To see this, first observe that $p_a - \Delta = p_b + (d_{ab} - \epsilon) \geq 0$ where the last inequality follows because $d_{ab} \geq \epsilon$. We then consider the three cases:

1. For $i \in S_1$, $p_i' = p_i - \Delta \geq p_a - \Delta \geq 0$ because $p_i \geq p_a$.

2. For $i \in S_2$, $p_i' = p_a - \Delta \geq 0$.

3. For $i \in S_3$, $p_i' = p_i \geq 0$.

Finally, we verify that for all $(i,j)$, $|p_i' - p_j'| \leq d_{ij}'$. First, observe that $p_a' - p_b' = (p_b + (d_{ab} - \epsilon)) - p_b' = d_{ab} - \epsilon = d_{ab}'$, and so the inequality is satisfied for index pair $(a,b)$. For all the other pairs $(i,j) \neq (a,b)$, we have $d_{ij}' = d_{ij}$, so it is enough to show that $|p_i' - p_j'| \leq d_{ij}$. Note that for all $x, y \in \{1, 2, 3\}$ with $x < y$, if $i \in S_x$ and $j \in S_y$, we have that $x \leq y$. Therefore, it is sufficient to verify the following six cases:

1. $i \in S_1, j \in S_1$: $|p_i' - p_j'| = (p_i - \Delta) - (p_j - \Delta) = p_i - p_j \leq d_{ij}$

2. $i \in S_1, j \in S_2$: $|p_i' - p_j'| = (p_i - \Delta) - (p_a - \Delta) = p_i - p_a < p_i - p_j \leq d_{ij}$

3. $i \in S_1, j \in S_3$: $|p_i' - p_j'| = (p_i - \Delta) - p_j = (p_i - p_j) - \Delta \leq (p_i - p_j) \leq d_{ij}$

4. $i \in S_2, j \in S_2$: $|p_i' - p_j'| = (p_a - \Delta) - (p_a - \Delta) = 0 \leq d_{ij}$

5. $i \in S_2, j \in S_3$: $|p_i' - p_j'| = (p_a - \Delta) - p_j \leq p_i - p_j \leq d_{ij}$

6. $i \in S_3, j \in S_3$: $|p_i' - p_j'| = p_i - p_j \leq d_{ij}$

Thus, we have shown that $p'$ is a feasible solution to $LP(r, d')$.

**Objective Value:** Note that for each index $i$, $p_i - p_i' \leq \Delta \leq \epsilon$. Therefore we have:

$$\langle r, \pi(r, d) \rangle - \langle r, \pi(r, d') \rangle \leq \langle r, \pi(r, d) \rangle - \langle r, p' \rangle$$
$$= \langle r, p - p' \rangle$$
$$\leq \epsilon \sum_{i=1}^{k} r_i$$

which completes the proof. $\square$

We now prove the main technical lemma of this section. It states that in any round in which the error of our distance estimates for *tight constraints* is small (even if we have high error in the distance estimates for slack constraints), then we will have low per-round regret.

**Lemma 4.** *At round $t$, if for all pairs of indices $(i,j)$, we have either:*

1. $|\hat{d}_{i,j}^t - \bar{d}_{i,j}^t| \leq \epsilon$ *or*

2. $v_{i,j}^t = 0$ *(corresponding to an LP constraint that is not tight)*

*then:*

$$\langle r^t, \pi(r^t, \bar{d}^t)\rangle - \langle r^t, \pi(r^t, \hat{d}^t)\rangle \le \epsilon k^3$$

*for any vector $r^t$ with $\|r^t\|_\infty \le 1$.*

*Proof.* First, define $\tilde{d}^t$ to be the coordinate-wise maximum of $\hat{d}^t$ and $\bar{d}^t$: i.e. the vector such that for every pair of coordinates $i, j$, $\tilde{d}_{ij} = \max(\bar{d}_{ij}, \hat{d}_{ij})$. To simplify notation, we will write $\hat{p} = \pi(r^t, \hat{d}^t)$, $\bar{p} = \pi(r^t, \bar{d}^t)$, and $\tilde{p} = \pi(r^t, \tilde{d}^t)$.

We make three relevant observations:

1. First, because $LP(r^t, \tilde{d}^t)$ is a relaxation of $LP(r^t, \bar{d}^t)$, it has only larger objective value. In other words, we have that $\langle r^t, \tilde{p}\rangle \ge \langle r^t, \bar{p}\rangle$. Thus, it suffices to prove that $\langle r^t, \hat{p}\rangle \ge \langle r^t, \tilde{p}\rangle - \epsilon k^3$.

2. Second, for all pairs $i, j$, $|\hat{d}_{i,j}^t - \tilde{d}_{i,j}^t| \le |\hat{d}_{i,j}^t - \bar{d}_{i,j}^t|$. Thus, if we had $|\hat{d}_{i,j}^t - \bar{d}_{i,j}^t| \le \epsilon$, we also have $|\hat{d}_{i,j}^t - \tilde{d}_{i,j}^t| \le \epsilon$.

3. Finally, by construction, for every pair $(i, j)$, we have $\tilde{d}_{ij} \ge \hat{d}_{ij}$

Let $S_1$ be the set of indices $(i, j)$ such that $|\hat{d}_{i,j}^t - \tilde{d}_{i,j}^t| \le \epsilon$, and let $S_2$ be the set of indices $(i, j) \notin S_1$ such that $v_{i,j}^t = 0$. Note that by assumption, these partition the space, and that by construction, for every $(i, j) \in S_2$, the corresponding constraint in $LP(r^t, \hat{d}^t)$ is not tight: i.e. $|\hat{p}_i - \hat{p}_j| < \hat{d}_{i,j}^t$. Let $d^*$ be the vector such that for all $(i, j) \in S_1$, $d_{ij}^* = \hat{d}_{ij}$, and for all $(i, j) \in S_2$, $d_{ij}^* = \tilde{d}_{ij}$. Observe that $LP(r^t, d^*)$ corresponds to a relaxation of $LP(r^t, \hat{d})$ in which *only constraints that were already slack were relaxed*. As a result, $\hat{p}$ is also an optimal solution to $LP(r^t, d^*)$. Note also that by construction, we now have that for *every* pair $(i, j)$: $|\tilde{d}_{ij} - d_{ij}^*| \le \epsilon$

Our argument will proceed by describing a sequence of $n + 1 = k^2 + 1$ vectors $p^0, p^1, \ldots, p^n$ such that $p^0 = \tilde{p}$, $p^n$ is a feasible solution to $LP(r^t, d^*)$, and for all adjacent pairs $p^\ell, p^{\ell+1}$, we have: $\langle r^t, p^{\ell+1}\rangle \ge \langle r^t, p^\ell\rangle - \epsilon k$. Telescoping these inequalities yields:

$$\langle r^t, \hat{p}\rangle \ge \langle r^t, p^n\rangle \ge \langle r^t, \tilde{p}\rangle - k^3 \epsilon$$

which will complete the proof.

To finish the argument, fix an arbitrary ordering on the indices $(i, j) \in [k] \times [k]$, which we denote by $(i_1, j_1), \ldots, (i_n, j_n)$. Define the distance vector $d^\ell$ such that:

$$d_{i_a, j_a}^\ell = \begin{cases} \tilde{d}_{i_a, j_a}, & \text{If } a > \ell; \\ d_{i_a, j_a}^*, & \text{If } a \le \ell. \end{cases}$$

Note that the sequence of distance vectors $d^1, \ldots, d^n$ "walks between" $\tilde{d}$ and $d^*$ one coordinate at a time. Now let $p^\ell = \pi(r^t, d^\ell)$. By construction, we have that every pair $(d^\ell, d^{\ell+1})$ differ in only a single coordinate, and that the difference has magnitude at most $\epsilon$. Therefore, we can apply Lemma 3 to conclude that:

$$\langle r^t, p^{\ell+1}\rangle \ge \langle r^t, p^\ell\rangle - \epsilon \sum_{i=1}^k r_i^t \ge \langle r^t, p^\ell\rangle - \epsilon k$$

as desired. $\qquad\square$

Finally, we have all the pieces we need to prove a regret bound for $L_{\text{known}-\theta}$.

**Theorem 2.** *For any time horizon T:*

$$\mathbf{Regret}(L_{known-\theta}, T) \le O\left(k^2 d^2 \log\left(\frac{d \cdot \|A^\top A\|_F}{\epsilon}\right) + k^3 \epsilon T\right)$$

*Setting $\epsilon = O(1/(k^3 T))$ yields a regret bound of $O(d^2 \log(\|A^\top A\|_F \cdot dkT))$.*

*Proof.* We partition the rounds $t$ into two types. Let $S_1$ denote the rounds such that there is at least one pair of indices $(i,j)$ such that one instance **DistanceEstimator**$_{ij}$ produced an estimate that had error more than $\epsilon$, and it was provided feedback. We let $S_2$ denote the remaining rounds, for which for *every* pair of indices $(i,j)$, *either* **DistanceEstimator**$_{ij}$ produced an estimate that had error at most $\epsilon$, or **DistanceEstimator**$_{ij}$ was not given feedback.

$$S_1 = \{t : \exists (i,j) : |\hat{d}_{ij}^t - \bar{d}_{ij}^t| > \epsilon \text{ and } v_{ij}^t = 1\} \quad S_2 = \{t : \forall (i,j) : |\hat{d}_{ij}^t - \bar{d}_{ij}^t| \le \epsilon \text{ or } v_{ij}^t = 0\}$$

Observe that $S_1$ and $S_2$ partition the set of all rounds. Next, observe that Corollary 1 tells us that:

$$|S_1| \le O\left(k^2 d^2 \log\left(\frac{d \cdot \|A^\top A\|_F}{\epsilon}\right)\right)$$

and Lemma 4 tells us that for every round $t \in S_2$, the per-round regret is at most $\epsilon k^3$. Together with the facts that $|S_2| \le T$ and that the per-round regret for any $t \in S_1$ is at most 1, we obtain:

$$\mathbf{Regret}(L_{\text{known}-\theta}, T) \le O\left(k^2 d^2 \log\left(\frac{d \cdot \|A^\top A\|_F}{\epsilon}\right) + k^3 \epsilon T\right)$$

$\square$

# 4 The Full Algorithm

In this section, we present our final algorithm, which has no knowledge of either the distance function $d$ or the linear objective $\theta$. The resulting algorithm shares many similarities with the algorithm we developed in Section 3, and so much of the analysis can be reused.

## 4.1 Outline of the Solution

At a high level, our plan will be to combine the techniques we developed in Section 3 with a standard "optimism in the face of uncertainty" strategy for learning the parameter vector $\theta$. Our algorithm will maintain a ridge-regression estimate $\tilde{\theta}$ together with confidence regions derived in Abbasi-Yadkori et al. [2011]. After it observes the contexts $x_i^t$ at round $t$, it uses these to derive upper confidence bounds on the expected rewards, corresponding to each context — represented as a vector $\hat{r}^t$. The algorithm continues to maintain distance estimates $\hat{d}^t$ using the **DistanceEstimator** subroutines, identically to how they were used in Section 3. At ever round, the algorithm then chooses its action according to the distribution $\pi^t = \pi(\hat{r}^t, \hat{d}^t)$.

The regret analysis of the algorithm follows by decomposing the per-round regret into two pieces. The first can be bounded by the sum of the *expected widths* of the confidence intervals

corresponding to each context $x_i^t$ that might be chosen at each round $t$, where the expectation is over the randomness of the algorithm's distribution $\pi^t$. A theorem of Abbasi-Yadkori et al. [2011] bounds the sum of the widths of the confidence intervals corresponding to arms *actually chosen* by the algorithm (Lemma 6). Using a martingale concentration inequality, we are able to relate these two quantities (Lemma 8). We show that the second piece of the regret bound can be manipulated into a form that can be bounded using Lemmas 1 and 4 from Section 3 (Theorem 3).

## 4.2   Confidence Intervals from Abbasi-Yadkori et al. [2011]

We would like to be able to construct confidence intervals at each round $t$ around each arm's expected reward such that for each arm $i$, with probability $1-\delta$, $\bar{r}_i^t \in [\tilde{r}_i^t + w_i^t, \tilde{r}_i^t + w_i^t]$, where $\tilde{r}_i^t$ is our ridge-regression estimate of $\bar{r}_i^t$ and $w_i^t$ is the confidence interval width around the estimate. Our algorithm will make use of such confidence intervals for the ridge regression estimator derived and analyzed in Abbasi-Yadkori et al. [2011], which we recount here.

Let $\tilde{V}^t = X^{t\top}X^t + \lambda I$ be a regularized design matrix, where $X^t = [x_{i_1}^1, \ldots, x_{i_{t-1}}^{t-1}]$ represents all the contexts whose rewards we have observed up to but not including time $t$. Let $Y^t = [r_{i_1}^1, \ldots, r_{i_{t-1}}^{t-1}]$ be the corresponding vector of observed rewards. $\tilde{\theta} = (V^t)^{-1}X^{t\top}Y^t$ is the (ridge regression) regularized least squares estimator we use at time $t$. We write $\tilde{r}_i^t = \langle \tilde{\theta}, x_i^t \rangle$ for the reward point prediction that this estimator makes at time $t$ for arm $i$.

We can construct the following confidence intervals around $\tilde{r}^t$:

**Lemma 5** (Abbasi-Yadkori et al. [2011]). *With probability $1 - \delta$,*

$$|\bar{r}_i^t - \tilde{r}_i^t| = |\langle x_i^t, (\theta - \tilde{\theta}) \rangle| \le \|x_i^t\|_{(\bar{V}^t)^{-1}} \left( \sqrt{2d \log(\frac{1 + t/\lambda}{\delta})} + \sqrt{\lambda} \right)$$

*where $\|x\|_A = \sqrt{x^\top A x}$*

Therefore, the confidence interval widths we use in our algorithm will be

$$w_i^t = \min(\|x_i^t\|_{(\bar{V}^t)^{-1}} \left( \sqrt{2d \log(\frac{1 + t/\lambda}{\delta})} + \sqrt{\lambda} \right), 1)$$

(expected rewards are bounded by 1 in our setting, and so the minimum maintains the validity of the confidence intervals). The upper confidence bounds we use to compute our distribution over arms will be $\hat{r}_i^t = \tilde{r}_i^t + w_i^t$. We will write $w^t = [w_1^t, \ldots, w_k^t]$ to denote the vector of confidence interval widths at round $t$.

Little can be said about the widths of these confidence intervals in isolation. However, the following theorem bounds the *sum* (over time) of the widths of the confidence intervals around the contexts actually selected.

**Lemma 6** (Abbasi-Yadkori et al. [2011]).

$$\sum_{t=1}^{T} w_{i^t}^t \le \sqrt{2d \log\left(1 + \frac{T}{d\lambda}\right)} \left( \sqrt{2dT \log(\frac{1 + T/\lambda}{\delta})} + \sqrt{T\lambda} \right)$$

**for** $i, j = 1, \ldots, k$ **do**

    $\mathbf{DistanceEstimator}_{ij} = \mathbf{DistanceEstimator}(\epsilon^2)$

**end**

**for** $t = 1, \ldots, T$ **do**

    receive the contexts $x^t = (x_1^t, \ldots, x_k^t)$

    $X^t = [x^1, \ldots, x^{t-1}]$

    $Y^t = [r^t, \ldots, r^{t-1}]$

    $\tilde{V}^t = {X^t}^\top X^t + \lambda I$

    $\tilde{\theta} = (V^t)^{-1} {X^t}^\top Y^t$

    **for** $i = 1, \ldots, k$ **do**

        $\tilde{r}_i^t = \langle \tilde{\theta}, x_i^t \rangle$

        $w_i^t = \min\left( \|x_i^t\|_{(\tilde{V}^t)^{-1}} \left( \sqrt{2d \log(\frac{1+t/\lambda}{\delta})} + \sqrt{\lambda} \right), 1 \right)$

        $\hat{r}_i^t = \tilde{r}_i^t + w_i^t$

    **end**

    **for** $i, j = 1, \ldots, k$ **do**

        $u_{i,j}^t = flatten((x_i^t - x_j^t)(x_i^t - x_j^t)^T))$

        $g_{i,j}^t = \mathbf{DistanceEstimator}_{i,j}.guess(u_{i,j}^t)$

        $\hat{d}_{ij}^t = \sqrt{g_{i,j}^t}$

    **end**

    $\pi^t = \pi(\hat{r}^t, \hat{d}^t)$

    Pull an arm $i^t$ according to $\pi^t$ and receive a reward $r_{i^t}^t$

    $S = O_d(x^t, \pi^t)$

    $R = \{(i,j) | (i,j) \notin S \wedge |\pi_i^t - \pi_j^t| = \hat{d}_{i,j}^t\}$

    **for** $(i,j) \in S$ **do**

        $\mathbf{DistanceEstimator}_{i,j}.feedback(\bot)$

        $v_{i,j}^t = 1$

    **end**

    **for** $(i,j) \in R$ **do**

        $\mathbf{DistanceEstimator}_{i,j}.feedback(\top)$

        $v_{i,j}^t = 1$

    **end**

**end**

**Algorithm 2:** $L_{\text{full}}$

## 4.3   The Algorithm

The pseudocode for the full algorithm is given in Algorithm 2.

In our proof of Theorem 3, we will connect the regret of $L_{full}$ to the sum of the *expected* widths of the confidence intervals pulled at each round. In contrast, what is bounded by Lemma 6 is the sum of the *realized* widths. Using the Azuma Hoeffding inequality, we can relate these two quantities.

**Lemma 7** (Azuma-Hoeffding inequality (Hoeffding [1963])). *Suppose* $\{X_k : k = 0, 1, 2, 3, \ldots\}$ *is a martingale and*

$$\left| X_k - X_{k-1} \right| < c_k.$$

*Then, for all positive integers N and all positive reals t,*

$$\Pr(X_N - X_0 \geq t) \leq \exp\left(\frac{t^2}{2 \sum_{k=1}^{N} c_k^2}\right)$$

**Lemma 8.**

$$\Pr\left( \sum_{t=1}^{T} \mathbb{E}_{i \sim \pi^t}[w_i^t] - \sum_{t=1}^{T} w_{i^t}^t \geq \sqrt{2T \log \frac{1}{\delta}} \right) \leq \delta$$

*Proof.* Once $x^1, \ldots, x^{t-1}, r_{i^t}^1, \ldots, r_{i^{t-1}}^{t-1}$ and $x^t$ are fixed, $\pi^t$ is fixed. In other words, for the filtration $\mathscr{F}^t = \sigma(x^1, \ldots, x^{t-1}, r_{i^t}^1, \ldots, r_{i^{t-1}}^{t-1}, x^t)$, $w_{i^t}^t$ is $\mathscr{F}^t$ measurable. Now, define

$$D^t = \sum_{s=1}^{t} \mathbb{E}_{i \sim \pi^s}[w_i^s] - \sum_{s=1}^{t} w_{i^s}^s$$

with respect to $\mathscr{F}^t$. One can think of $D^t$ as the accumulated difference between the confidence width of the arm that was actually pulled and the expected confidence width. It's easy to see that $\{D^t\}$ is a martingale, as $\mathbb{E}[D^1] = 0$, and $\mathbb{E}[D^{t+1}|\mathscr{F}^t] = D^t$.

Also, $D_t - D_{t-1} = w_{i^t}^t - \mathbb{E}_{i \sim \pi^t}[w_i^t] \leq 1$, since the confidence interval widths are bounded by 1.

Applying the Azuma-Hoeffding inequality gives us the following:

$$\Pr\left( \sum_{t=1}^{T} \mathbb{E}_{i \sim \pi^t}[w_i^t] - \sum_{t=1}^{T} w_{i^t}^t \geq \epsilon \right) = \Pr(D^T \geq \epsilon) \leq \exp\left(\frac{-\epsilon^2}{2T}\right)$$

Now, setting $\epsilon = \sqrt{2T \ln \frac{1}{\delta}}$ yields:

$$\Pr\left( \sum_{t=1}^{T} \mathbb{E}_{i \sim \pi^t}[w_i^t] - \sum_{t=1}^{T} w_{i^t}^t \geq \sqrt{2T \log \frac{1}{\delta}} \right) \leq \delta$$

$\square$

**Theorem 3.** *For any time horizon T, with probability* $1 - \delta$:

$$\mathbf{Regret}(L_{full}, T) \leq O\left( k^2 d^2 \log\left( \frac{d \cdot \|A^\top A\|_F}{\epsilon} \right) + k^3 \epsilon T + d\sqrt{T} \log(\frac{T}{\delta}) \right)$$

*If* $\epsilon = 1/k^3 T$, *this is a regret bound of* $O\left( k^2 d^2 \log\left( kdT \cdot \|A^\top A\|_F \right) + d\sqrt{T} \log(\frac{T}{\delta}) \right)$

*Proof.* We can compute:

$$\mathbf{Regret}(L_{full}, T) = \sum_{t=1}^{T} \mathop{\mathbb{E}}_{i \sim \pi(\bar{r}^t, \bar{d}^t)} [\bar{r}_i^t] - \sum_{t=1}^{T} \mathop{\mathbb{E}}_{i \sim \pi(\hat{r}^t, \hat{d}^t)} [\bar{r}_i^t]$$

$$= \sum_{t=1}^{T} \langle \bar{r}^t, \pi(\bar{r}^t, \bar{d}^t) \rangle - \langle \bar{r}^t, \pi(\hat{r}^t, \hat{d}^t) \rangle$$

$$= \sum_{t=1}^{T} \langle \bar{r}^t, \pi(\bar{r}^t, \bar{d}^t) \rangle - \langle \bar{r}^t, \pi(\hat{r}^t, \bar{d}^t) \rangle + \langle \bar{r}^t, \pi(\hat{r}^t, \bar{d}^t) \rangle - \langle \bar{r}^t, \pi(\hat{r}^t, \hat{d}^t) \rangle$$

$$\leq \sum_{t=1}^{T} \langle \hat{r}^t, \pi(\hat{r}^t, \bar{d}^t) \rangle - \langle \bar{r}^t, \pi(\hat{r}^t, \bar{d}^t) \rangle + \langle \bar{r}^t, \pi(\hat{r}^t, \bar{d}^t) \rangle - \langle \bar{r}^t, \pi(\hat{r}^t, \hat{d}^t) \rangle$$

$$\leq \sum_{t=1}^{T} \langle 2w^t, \pi(\hat{r}^t, \bar{d}^t) \rangle + \langle \bar{r}^t, \pi(\hat{r}^t, \bar{d}^t) \rangle - \langle \bar{r}^t, \pi(\hat{r}^t, \hat{d}^t) \rangle$$

Here, the first inequality follows from the fact that $\hat{r}^t$ is coordinate-wise larger than $\bar{r}^t$, and that $\pi(\hat{r}^t, \bar{d}^t)$ is the optimal solution to $LP(\hat{r}^t, \bar{d}^t)$. The second inequality follows from $\bar{r} \in [\tilde{r} - w, \tilde{r} + w] = [\hat{r} - 2w, \hat{r}]$.

Just as in the proof of Theorem 2, we now partition time into two sets:

$$S_1 = \{t : \exists (i,j) : |\hat{d}_{ij}^t - \bar{d}_{ij}^t| > \epsilon \text{ and } v_{ij}^t = 1\} \quad S_2 = \{t : \forall (i,j) : |\hat{d}_{ij}^t - \bar{d}_{ij}^t| \leq \epsilon \text{ or } v_{ij}^t = 0\}$$

Recall that corollary 1 bounds $|S_1| \leq O\left(k^2 d^2 \log\left(\frac{d \cdot \|A^\top A\|_F}{\epsilon}\right)\right)$. Since the per-step regret of our algorithm can be at most 1, this means that rounds $t \in S_1$ can contribute in total at most $C \doteq O\left(k^2 d^2 \log\left(\frac{d \cdot \|A^\top A\|_F}{\epsilon}\right)\right)$ regret. Thus, for the rest of our analysis, we can focus on rounds $t \in S_2$.

Fix any round $t \in S_2$. From Lemma 4 we have:.

$$\langle \hat{r}, \pi(\hat{r}, \bar{d}) \rangle - \langle \hat{r}, \pi(\hat{r}, \hat{d}) \rangle \leq k^3 \epsilon$$

Further manipulations give:

$$\left( \langle \hat{r}, \pi(\hat{r}, \bar{d}) \rangle - \langle \bar{r}, \pi(\hat{r}, \bar{d}) \rangle \right) - \left( \langle \hat{r}, \pi(\hat{r}, \hat{d}) \rangle - \langle \bar{r}, \pi(\hat{r}, \hat{d}) \rangle \right) \leq k^3 \epsilon - \langle \bar{r}, \pi(\hat{r}, \bar{d}) \rangle + \langle \bar{r}, \pi(\hat{r}, \hat{d}) \rangle$$

$$\langle 2w, \pi(\hat{r}, \bar{d}) \rangle - \langle 2w, \pi(\hat{r}, \hat{d}) \rangle \leq k^3 \epsilon - \langle \bar{r}, \pi(\hat{r}, d) \rangle + \langle \bar{r}, \pi(\hat{r}, \hat{d}) \rangle$$

$$\langle 2w, \pi(\hat{r}, \bar{d}) \rangle \leq \langle 2w, \pi(\hat{r}, \hat{d}) \rangle + k^3 \epsilon - \langle \bar{r}, \pi(\hat{r}, \bar{d}) \rangle + \langle \bar{r}, \pi(\hat{r}, \hat{d}) \rangle$$

Now, substituting the above expressions back into our expression for regret:

**Regret**$(L_{full}, T)$

$$\leq C + \sum_{t \in S_2} \langle 2w^t, \pi(\hat{r}^t, \bar{d}^t) \rangle + \langle \bar{r}^t, \pi(\hat{r}^t, \bar{d}^t) \rangle - \langle \bar{r}_i^t, \pi(\hat{r}^t, \hat{d}^t) \rangle$$

$$\leq C + \sum_{t \in S_2} \langle 2w^t, \pi(\hat{r}^t, \hat{d}^t) \rangle + k^3 \epsilon - \langle \bar{r}^t, \pi(\hat{r}^t, \bar{d}^t) \rangle + \langle \bar{r}^t, \pi(\hat{r}^t, \hat{d}^t) \rangle + \langle \bar{r}^t, \pi(\hat{r}^t, \bar{d}^t) \rangle - \langle \bar{r}_i^t, \pi(\hat{r}^t, \hat{d}^t) \rangle$$

$$\leq C + \sum_{t \in S_2} \langle 2w^t, \pi(\hat{r}^t, \hat{d}^t) \rangle + k^3 \epsilon$$

$$\leq C + 2 \sum_{t \in S_2} \mathbb{E}_{i \in \pi(\hat{r}^t, \hat{d}^t)} [w_i^t] + k^3 \epsilon$$

$$\leq C + k^3 \epsilon T + 2 \left( \sqrt{2d \log\left(1 + \frac{T}{d\lambda}\right)} \left( \sqrt{2dT \log(\frac{1 + T/\lambda}{\delta})} + \sqrt{T\lambda} \right) + \sqrt{2T \log \frac{1}{\delta}} \right)$$

$$= O\left( k^2 d^2 \log\left( \frac{d \cdot \|A^\top A\|_F}{\epsilon} \right) \right) + k^3 \epsilon T + O(d\sqrt{T} \log(\frac{T}{\delta}))$$

The last inequality holds with probability $1 - \delta$ and uses Lemmas 6 and 8, and sets $\lambda = 1$.

$\square$

Finally, the bound on the fairness loss is identical to the bound we proved in Theorem 1 (because our algorithm for constructing distance estimates $\hat{d}$ is unchanged). We have:

**Theorem 4.** *For any sequence of contexts and any Mahalanobis distance* $d(x_1, x_2) = \|Ax_1 - Ax_2\|_2$:

$$\textbf{FairnessLoss}(L_{full}, T, \epsilon) \leq O\left( k^2 d^2 \log\left( \frac{d \cdot \|A^\top A\|_F}{\epsilon} \right) \right)$$

## 5  Conclusion and Future Directions

We have initiated the study of fair sequential decision making in settings where the notions of payoff and fairness are separate and may be in tension with each other, and have shown that in a stylized setting, optimal fair decisions can be efficiently learned *even without direct knowledge of the fairness metric*. A number of extensions of our framework and results would be interesting to examine. At a high level, the interesting question is: how much can we further relax the information about the fairness metric available to the algorithm? For instance, what if the fairness feedback is only partial, identifying some but not all fairness violations? What if it only indicates whether or not there were any violations, but does not identify them? What if the feedback is not guaranteed to be exactly consistent with any metric? Or what if the feedback is consistent with *some* distance function, but not one in a known class: for example, what if the distance is not exactly Mahalanobis, but is approximately so? In general, it is very interesting to continue to push to close the wide gap between the study of individual fairness notions and the study of group fairness notions. When can we obtain the strong semantics of individual fairness without making correspondingly strong assumptions?

# Acknowledgements

We thank Steven Wu and Matthew Joseph for helpful discussions at an early stage of this work.

## Footnotes

[1] A random variable $X$ with $\mu = \mathbb{E}[X]$ is sub-gaussian, if for all $t \in \mathbb{R}$, $\mathbb{E}[e^{t(X-\mu)}] \leq e^{\frac{t^2}{2}}$.

[2] We assume that $h^{T+1}$ is generated by algorithm $A$, meaning randomness only comes from the stochastic reward and the way in which each arm is selected according to the probability distribution calculated by the algorithm. We don't assume any distributional assumption over $x^1,\dots,x^T$

[3]If the algorithm also learned whether or not its guess was in error by more than $\epsilon$ at each round, variants of the classical halving algorithm could obtain this guarantee. But the algorithm does not receive this feedback, which is why the more sophisticated algorithm of Lobel et al. [2017] is needed.

[4]In Lobel et al. [2017], the algorithm receives feedback in every round, and the scale parameters $B_1$ and $B_2$ are normalized to be 1. But the version we state is an immediate consequence.

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

## A   Generalization to Multiple Actions

In the body of the paper, we analyzed the standard contextual bandit setting in which the algorithm must choose a *single* action to take at each round. However, it is often the case that this constraint is artificial and undesirable in settings for which fairness is a concern. Consider, for example, the case of lending: at each round, a bank observes the loan applications of a collection of individuals, and decides whom to grant loans to. Some loans may be profitable and some loans may not be — so the optimal policy is non-trivial. But there need not be a budget constraint — the optimal policy may grant loans to as many qualified individuals as there are on a given day.

In our framework, this corresponds to letting the algorithm take as many as $k$ actions on a single day. Fortunately, all of our results generalize to this case. The maximum reward per day in this case increases from 1 to $k$, so naturally the regret bound we obtain is also a factor of $k$ larger. In this section, we explain the details of our proof that need to be modified.

The first step is to consider a modified linear program $LP(a,c)$, which we will write as $LP_m(a,c)$. It simply replaces the simplex constraint that the probabilities of actions sum to 1 with the hypercube constraint that no probability can be greater than 1:

$$\underset{\pi=\{p_1,\dots,p_k\}}{\text{maximize}} \quad \sum_{i=1}^{k} p_i a_i$$
$$\text{subject to} \quad |p_i - p_j| \leq c_{i,j}, \forall (i,j)$$
$$0 \leq p_i \leq 1, \forall i$$

We must also change our definition of regret, because the benchmark we want to compete with is the best fair policy that can make up to $k$ action selections per round. This simply corresponds to comparing to a benchmark which is defined with respect to $LP_m(a,c)$ — but the form of the regret is unchanged:

**Regret**$_m(L, T)$

$$= \sum_{t=1}^{T} \sum_{i=1}^{k} \bar{r}_i^t \cdot Pr(\text{best fair policy pulls arm } i \text{ in round } t) - \bar{r}_i^t \cdot Pr(L \text{ pulls arm } i \text{ in round } t)$$

$$= \sum_{t=1}^{T} \langle \bar{r}^t, \pi(\bar{r}^t, \bar{d}^t) \rangle - \langle \bar{r}^t, f^t(h^t, x^t) \rangle$$

where $\pi$ is defined exactly as before, except with respect to $LP_m(a,c)$.

The first observation is that our generalization to multiple arms does not affect our analysis of fairness loss at all, since we are able to bound this without reference to the rewards. That is, we still have that fairness loss is bounded as

$$\textbf{FairnessLoss}(L_{full_m}, T, \epsilon) \leq O\left( k^2 d^2 \log\left( \frac{d \cdot \|A^\top A\|_F}{\epsilon} \right) \right)$$

As for our regret analysis, certain terms in the regret scale by a factor of $k$.

$$\textbf{Regret}_m(L_{full_m}, T) \leq O\left( k^3 d^2 \log\left( \frac{d \cdot \|A^\top A\|_F}{\epsilon} \right) + k^3 \epsilon T + dk\sqrt{k^2 T} \log(\frac{kT}{\delta}) \right)$$

*Proof.* There are only two parts of our proof that depend on the structure on the linear program $LP(a,c)$. The first is the proof of Lemma 3, which uses the fact that if we take a feasible solution to $LP(a,c)$ and reduce its values pointwise, we maintain feasibility — that is, that the feasible region of $LP(a,c)$ is downward closed. But note that the feasible region of $LP_m(a,c)$ is also downward closed, so the same argument goes through. Recall that our analysis in the known objective case partitions rounds into two sorts: rounds for which we can bound our per-round regret (from

Lemma [3]), and a bounded number of rounds in which we cannot. For those rounds in which we cannot bound the per-round regret, the maximum regret is now $k$ rather than 1. So, our regret during these rounds increases by a factor of $k$ to $O\left(k^3 d^2 \log\left(\frac{d \cdot \|A^\top A\|_F}{\epsilon}\right)\right)$.

Therefore, we have that

$$\mathbf{Regret}_m(L_{full_m}, T) \le O\left(k^3 d^2 \log\left(\frac{d \cdot \|A^\top A\|_F}{\epsilon}\right)\right) + k^3 \epsilon T + \sum_{t \in S_2} \langle 2w^t, \pi(\hat{r}^t, \hat{d}^t)\rangle$$

where $S_2 = \{t : \forall (i,j) : |\hat{d}_{ij}^t - \bar{d}_{ij}^t| \le \epsilon \text{ or } v_{ij}^t = 0\}$

Next, we need to consider the final term in this expression. $\langle w^t, \pi(\hat{r}^t, \hat{d}^t)\rangle$ is the expected sum of the confidence interval widths of the arms that are pulled at round $t$. By the same martingale argument as in lemma [8], with high probability, the expected sum of the confidence interval widths over time horizon $T$ is close to the realized sum of the confidence widths of the arms pulled; in this case, the martingale is

$$D^t = \sum_{s=1}^{t} \sum_{i=1}^{k} w_i^s \cdot \Pr(\text{arm } i \text{ is pulled in round } s) - \sum_{s=1}^{t} \sum_{i=1}^{k} w_i^s \cdot \mathbb{1}(\text{arm } i \text{ is pulled in round } s)$$

However, in this case, the martingale difference is bounded by at most $k$ instead of 1. Hence, applying the Azuma-Hoeffding inequality gives us that with probability $1 - \delta$,

$$\sum_{t=1}^{T} \sum_{i=1}^{k} w_i^t \cdot \Pr(\text{arm } i \text{ is pulled in round } t) \le \sum_{t=1}^{T} \sum_{i=1}^{k} w_i^t \cdot \mathbb{1}(\text{arm } i \text{ is pulled in round } t) + \sqrt{2k^2 T \log\frac{1}{\delta}}$$

First, note that the confidence interval derived from lemma [5] remains valid. Also, $\bar{V}^t = \bar{V}^{t-1} + \sum_{i \in P^t} x_i^t x_i^{t\top}$. For simplicity in notation, we write $P^t = \{i : \text{arm } i \text{ is pulled in round } t\}$. So we need to bound $\sum_{t=1}^{T} \sum_{i \in P^t} w_i^t$.

We can then derive:

$$\sum_{t=1}^{T} \sum_{i \in P^t} w_i^t \le \sum_{t=1}^{T} \sum_{i \in P^t} \|x_i^t\|_{(\bar{V}^{t-1})^{-1}} \left(\sqrt{2d \log(\frac{1+t/\lambda}{\delta})} + \sqrt{\lambda}\right)$$

$$\le \sum_{t=1}^{T} \sum_{i \in P^t} \|x_i^t\|_{(\bar{V}^{t-1})^{-1}} \left(\sqrt{2d \log(\frac{1+t/\lambda}{\delta})}\right) + \sum_{t=1}^{T} \sum_{i \in P^t} \left(\|x_i^t\|_{(\bar{V}^{t-1})^{-1}} \sqrt{\lambda}\right)$$

$$\le \sum_{t=1}^{T} \sum_{i \in P^t} \|x_i^t\|_{(\bar{V}^{t-1})^{-1}} \cdot \left(\sqrt{\sum_{t=1}^{T} \sum_{i \in P^t} 2d \log(\frac{1+t/\lambda}{\delta})}\right) + \sqrt{\sum_{t=1}^{T} \sum_{i \in P^t} \lambda}$$

$$\le \sum_{t=1}^{T} \sum_{i \in P^t} \|x_i^t\|_{(\bar{V}^{t-1})^{-1}} \cdot \left(\sqrt{2dkT \log(\frac{1+kT/\lambda}{\delta})}\right) + \sqrt{kT\lambda}$$

For each $i \in [k]$, write $A_i$ to denote the set of rounds that arm $i$ is pulled. $\sum_{t=1}^{T} \sum_{i \in P^t} \|x_i^t\|_{(\bar{V}^t)^{-1}} = \sum_{i=1}^{k} \sum_{t \in A_i} \|x_i^t\|_{(\bar{V}^t)^{-1}}$, so for each $i \in [k]$, we'll bound $\sum_{t \in A_i} \|x_i^t\|_{(\bar{V}^t)^{-1}}$.

**Lemma 9.**

$$\sum_{t \in A_i} \|x_i^t\|_{(V^{t-1})^{-1}} \le \sqrt{2d \log\left(1 + \frac{kT}{d\lambda}\right)}$$

*Proof.* We'll iterate each $\|x_i^t\|_{(V^{t-1})^{-1}}$ first over round $t = 1, \ldots, T$ and then $j \in P^t$ where the order of $P^t$ has its very first element as $\|x_i^t\|$ and the rest is arbitrary. Let's call this indexing $a$. First, we have that $\bar{V}(a) = \bar{V}(a-1) + x(a)x(a)^\top$. More importantly, because of the way we chose to index, for each $t \in A_i$ and index $a$ that corresponds to $(i, t)$, $\|x_i^t\|_{(\bar{V}^{t-1})^{-1}} = \|x(a)\|_{(\bar{V}(a-1))^{-1}}$

From Lemma 11 in Abbasi-Yadkori et al. [2011] we have $\sum_{a=1}^{N} \|x(a)\|_{(\bar{V}(a-1))^{-1}} \le \sqrt{2d \log\left(1 + \frac{N}{d\lambda}\right)}$, where $N \le kT$.

Therefore, we have that

$$\sum_{t \in A_i} \|x_i^t\|_{(V^{t-1})^{-1}} \le \sum_{a=1}^{N} \|x(a)\|_{(\bar{V}(a-1))^{-1}} \le \sqrt{2d \log\left(1 + \frac{kT}{d\lambda}\right)}$$

$\square$

Applying the above lemma for each arm $i \in [k]$, we have

$$\sum_{t=1}^{T} \sum_{i \in P^t} w_i^t \le \sum_{t=1}^{T} \sum_{i \in P^t} \|x_i^t\|_{(V^{t-1})^{-1}} \cdot \left(\sqrt{2dkT \log(\frac{1 + kT/\lambda}{\delta})}\right) + \sqrt{kT\lambda}$$

$$\le k\sqrt{2d \log\left(1 + \frac{kT}{d\lambda}\right)} \cdot \left(\sqrt{2dkT \log(\frac{1 + kT/\lambda}{\delta})}\right) + \sqrt{kT\lambda}$$

$\square$