[Reviews · NeurIPS 2018]

Reviewer 1



In this paper, the authors provide an algorithm for learning a fair contextual bandit model while learning the distance function inherent in individual fairness. This paper is primarily interesting in that specifying the distance metric is a core difficulty in making use of individual fairness. By enabling the distance metric to be learned through feedback as the model learns presents a way forward for individual fairness. However, in order to do this, the paper needs to make strong assumptions for the theory to work: the distance metric is a Mahalanobis metric (a possibly reasonable assumption) and a linear model (a much more restrictive assumption). Because of these assumptions, I doubt that this work will be useful in practice in the near future. However, the theoretical results I believe are valuable as a step in the right direction for making individual fairness practical.

Reviewer 2



The authors study the problem of correcting for individual fairness in an adversarial online learning problem when the fairness constraints depend on an unknown similarity metric between the items. This captures the setting in which a quantitative fairness metric over individuals cannot be pre-specified, but is clear a posteriori. The problem is crucial as it the setting of unknown similarities is extremely natural, and keeping the objective separate from the optimization goal seems often appropriate. They show that optimal fair decisions can be efficiently learned, even in this setting that does not have the direct knowledge of the fairness metric. The setting and approach seem appropriate, and the paper opens up many interesting directions for future work. The discussion of the related work in the supplementary material was great and I think crucial to situate this work -- I hope room can be made in the final version for it in the main body. To this I would just add two additional papers which also deal with different notions of fairness in online learning settings (https://arxiv.org/abs/1707.01875 and https://arxiv.org/abs/1802.08674).

Reviewer 3



The paper revisits the individual fairness definition from Dwork et al. TCS 2012, which (informally) requires similar individuals to be treated similarly (and assumes that the similarity metric is given), and presents an analysis how such metric can be learned in online learning settings. The authors consider the Mahalanobis metric to be learnt in linear contextual bandit settings with weak feedback, i.e. an oracle identifies fairness violations, but does not quantify them. In authors belief this represent the interventions of a regulator who “knows unfairness when he sees it”. The authors provide an interesting theoretical analysis and a novel algorithm in the adversarial context setting. I find the work application-inspired, but unfortunately not convincing. The presentation can be improved by providing a realistic example of how a regulator provides weak feedback. - I read the authors' response.